# Presence of *Mycobacterium avium* Subspecies *paratuberculosis* Monitored Over Varying Temporal and Spatial Scales in River Catchments: Persistent Routes for Human Exposure

**DOI:** 10.3390/microorganisms7050136

**Published:** 2019-05-15

**Authors:** Hollian Richardson, Glenn Rhodes, Peter Henrys, Luigi Sedda, Andrew J. Weightman, Roger W. Pickup

**Affiliations:** 1Biomedical and Life Sciences Division, Faculty of Health and Medicine, Lancaster University, Lancaster LA1 4YQ, UK; hollianrichardson88@hotmail.co.uk; 2Division of Molecular and Clinical Medicine, University of Dundee, Mailbox 12, Ninewells Hospital and Medical School, Dundee DD1 9SY, UK; 3Lake Ecosystems Group, Centre for Ecology and Hydrology, Lancaster Environment Centre, Lancaster LA1 4AP, UK; Glenn@ceh.ac.uk (G.R.); pehn@ceh.ac.uk (P.H.); 4Centre for Health Informatics, Computing and Statistics (CHICAS), Lancaster Medical School, Faculty of Health and Medicine, Lancaster University, Lancaster LA1 4YQ, UK; l.sedda@lancaster.ac.uk; 5School of Biosciences, Cardiff University, Sir Martin Evans Building, Museum Avenue, Cardiff CF10 3AX, UK; weightman@cardiff.ac.uk

**Keywords:** *Mycobacterium avium* subspecies *paratuberculosis*, freshwater environment, Crohn’s disease, catchment

## Abstract

*Mycobacterium avium* subspecies *paratuberculosis* (*Map*) was monitored by quantitative PCR over a range of temporal and spatial scales in the River Tywi catchment. This study shows the persistence of *Map* over a 10-year period with little change, which correlates with the recognised levels of Johne’s disease in British herds over that period (aim 1). *Map* was quantified within the river at up to 10^8^ cell equivalents L^−1^ and was shown to be consistently present when monitored over finer timescales (aim 4). Small wastewater treatment plants where the ingress of human-associated *Map* might be expected had no significant effect (aim 2). *Map* was found for the first time to be located in natural river foams providing another route for spread via aerosols (aim 5). This study provides evidence for the environmental continuum of *Map* from the grazing infected animal via rain driven runoff through field drains and streams into main rivers; with detection at a high frequency throughout the year. Should *Map* need to be monitored in the future, we recommend that weekly or monthly sampling from a fixed location on a river will capture an adequate representation of the flow dynamics of *Map* in a catchment (aim 3). The human exposure to *Map* during this process and its impact on human health remains unquantified.

## 1. Introduction

*Mycobacterium avium* subspecies *paratuberculosis* (*Map*) is a very slow growing member of the *Mycobacterium avium* complex [1,2]. *Map* is recognised as a multi-host pathogen [3] and is now implicated in a wide range of diseases including Parkinson’s disease, rheumatoid arthritis, neuromyelitis optica spectrum disorder, multiple sclerosis and type 1 diabetes myelitis [4,5,6,7,8,9,10,11,12]. However, it is known to cause Johne’s disease which is a chronic inflammation of the intestine (JD) [3,13,14,15], which can affect many animal species including primates [3]. This enteric animal pathogen is also significantly associated with Crohn’s disease (CD), which also involves a chronic inflammation of the intestine in humans [12,16,17,18]. This association remains controversial. Appropriate laboratory methods have shown that most people with chronic inflammation of the intestine of the Crohn’s disease type are infected with this chronic enteric pathogen [19,20,21,22,23,24]. Like that of Johne’s disease, the incidence of Crohn’s disease continues to increase with consequential economic health costs [25,26,27,28,29], particularly in children. Both Northern and Central Europe and Australia showed an average 5-fold per decade increase in the disease in children under 16 [30,31,32].

*M. avium* subsp. *paratuberculosis* can exist in animals for long periods (years) without causing clinical disease. Johne’s disease has now spread worldwide [33,34], and is particularly prevalent in Europe and North America [34]. Subclinical infection is significant in domestic livestock, especially in cattle, sheep and goats [14]. It is estimated that the herd prevalence for JD in cattle in the USA is 68% [35] and is 34.7% in U.K. [36,37,38,39].

Both clinically and sub-clinically infected animals can shed *Map* in the faeces onto grass land and pastures. The numbers shed depends on the species of the animal, the *Map* strain involved and the severity of the disease [40]. Once in the environment, *Map* can survive for many months/years in agricultural slurry and in the wider environment [41,42,43]. *Map* is washed by rainfall from contaminated pastures via surface waters into rivers [41,43]. Previously, we showed that for *Map* in the River Taff (Cardiff, South Wales, UK), its presence was almost predictable from rainfall patterns and river flow [43] and it was present in 32% of the samples taken. This contrasts with the River Tywi, where its presence in 69% of the samples was predictable based on rainfall and river height parameters. [41]. Furthermore, as a consequence of the endemic presence of *Map* in cattle in the Taff/Tywi catchments, deposition and transport from the catchments was so extensive that *Map* was maintained in the river for periods of several weeks at a time [43]. Both of these studies were qualitative and recognised the need for quantitative data [41,43].

Previously, we modelled the main human exposure routes of *Map* and suggested that its presence and distribution was primarily driven by shedding from clinically and sub-clinically infected animals, but other factors, such as slurrying and soil redistribution from water treatment, that recycle *Map* back from the catchment to the river, also influenced its presence and distribution in the environment [41,43]. Water from these catchments is used for abstraction and public supply. We detected *Map* by aerosol samples collected above the River Taff. In addition, we detected *Map* in domestic showers from different regions in the U.K. The detection of *Map* in river aerosols and those from domestic showers, provided the first evidence that aerosols are an exposure route for *Map* to humans and may play a role in the epidemiology of CD [44].

Furthermore, in the first comprehensive geographical survey of its distribution in Great Britain [45], it was shown that the presence of *Map* significantly increased from North to South of Great Britain. Its presence was significantly associated with increasing numbers of cattle over the same longitudinal axis. Assessment of samples from differing land usages showed that *Map* was widely distributed in both farming and non-farming areas. We showed that *Map* is widely distributed within and outside the confines of the farming environment; its geographical distribution was wider than originally anticipated. However, we concluded that monitoring rivers better describes the *Map* status of catchment than individual soil samples [45].

In this study, we returned to the River Tywi catchment in Wales, UK [41], with the following aims:to determine whether the overall distribution and presence of *Map* in the catchment has changed over time when farming practices and management of stock remain the same,to test whether waste water treatment effluent is a significant input of *Map* in the river,to assess whether presently used sampling regimes to detect *Map* in the environment underestimate its prevalence,to quantify the presence of *Map* in rivers, andto extend our catchment model [41].

Furthermore, we complemented our geographical study of *Map* distribution in the UK and Countryside Survey 2007 [45] by examining water samples from drains and steams around agricultural fields for *Map* from the River Eden catchment (England, UK), thus providing a sampling continuum from the clinically and sub-clinically infected cattle on the pasture to the estuaries which has provided new data for our original *Map* human exposure model [41].

## 2. Materials and Methods

### 2.1. Study Areas

#### 2.1.1. River Tywi, Wales, UK

The River Tywi is a major river in the South Wales region. Within its length of 75 miles, it has numerous tributaries including Cothi and Dulas with water fed from catchments of 302 km^2^ and 102 km^2^ respectively. It passes through areas of both cattle and sheep farming. The Tywi catchment is predominantly rural and comprises 1100 km^2^ containing 3188 agricultural units. Animal stocks comprise 108,142 mixed dairy and beef cattle and calves and 1,388,303 sheep and lambs, including 717,283 breeding ewes. River Tywi is subjected to water abstraction and is a major source of domestic supply. The source of the River Tywi lies in the afforested undulating moor land of the Cambrian Mountains of Mid Wales [41]. Llyn Brianne reservoir marks the start of the River Tywi, through a continual but flow-regulated release of water. The river falls steeply for 10 miles before it enters a flood plain near Llandovery. It meanders across a 1-mile-wide plain for 30 miles in a south-westerly direction to Carmarthe. It then enters a 12-mile-long tidal section which then enters the Bristol Channel in a broad estuary. Sample site 7 (Figure 1) represents the confluence of the Cothi and Tywi.

#### 2.1.2. Eden Catchment, Cumbria, UK

The River Eden is located in Eden Valley, Cumbria, and encompasses a catchment area of 2288 km^2^ [46]. The river forms at the convergence of two streams, the Red Gill and the Little Grain merge to form Hell Gill Beck [46]. The river flows past Hell Gill Force to become the River Eden and then flows into the Irish Sea via the Solway Firth (Owen et al., [46]). Located in the Eden catchment are three 10-km^2^ sub-catchments: the Darce sub-catchment, the Morland sub-catchment and the Pow sub-catchment [46]. The Darce sub-catchment is located to the West in the Eden catchment and is less than 5 km from Ullswater. This catchment is representative of the upland of the Eden catchment and is used for improved and rough grazing by livestock including sheep and cattle [46]. The Morland sub-catchment is located in the east of the Eden catchment and is used for both dairy farming and meat production [46]. The Pow sub-catchment is located 4 km south of Carlisle and, compared to the other two sub-catchments, is intensively farmed with diary, beef, sheep, pig and poultry.

#### 2.1.3. Sampling Sites

Water, sediment and foam samples were taken from sites in the Tywi catchment (Wales, UK, comprising rivers Tywi, Cothi and Dulas) and from field drains in the Eden catchment, Cumbria, UK. Sampling comprised 3 regimes; firstly, weekly and monthly samples from the rivers Tywi and Cothi (Table 1 and Figure 1). Using a fresh disposable bucket, 10-L water samples were taken from the middle of the river. Samples were sealed in disposable screw-cap plastic containers and sent by fast post to Lancaster University where they were stored at 4 °C prior to processing. Secondly, hourly samples over a 6-h period were taken on 3 occasions from River Tywi with samples also taken from the rivers Cothi and Dulas tributaries and at their confluence (Nantgaredig Bridge) on each occasion (Table 2 and Figure 1) and transported on ice before processing; and, thirdly a number of samples were taken from field drains in the River Eden sub-catchments of Pow, Darce and Morland (Cumbria, UK; Table 3). Natural foam [47] from river water was collected in sterile bottle before processing.

### 2.2. Sample Processing

One litre of river water was filtered using a vacuum filtering unit with a 0.2-µm pore size polycarbonate filter (Supor-200 Pall Life Sciences, Portsmouth, UK). The filter was removed aseptically and placed into a sterile McCartney bottle containing 2 mL Tris-HCl (0.1 M) (Sigma Aldrich Ltd., Gillingham, UK) and cells re-suspended by vortexing (5 min at full power). This sample was divided equally between two sterile 1.5-mL microcentrifuge tubes (Anachem Ltd., Leicester, UK) and centrifuged for 20 min at 13,000× *g*. The supernatant from one of the tubes was discarded while in the second, 800 μL supernatant was removed and the pellet was resuspended in the remaining 200 µL supernatant. Once mixed thoroughly, this was added to the first pellet and resuspended. Foam samples were resuspended in 1 mL sterile water and added to a bead beating tube. These samples then underwent DNA extraction as described previously [44].

### 2.3. Detection of Map by PCR

*Map* was detected by qPCR using single-tube duplex reactions that combined the previously described assays [48,49] to amplify *Map*-specific regions IS*900* and F57 respectively as described previously [44]. Standard curves, constructed in accordance with methods previously outlined, were used to quantify DNA [50] based on a genome size of 4.83 Mbp for *M. avium* ssp. *paratuberculosis* K-10 (GenBank Accession No. NC002944) and an average number of gene copies per genome of 17 (IS*900*) and 1 (F57) [51,52]. *Map* was quantified and represented as cell equivalents per litre (CE L^−1)^ [50].

### 2.4. Statistical Analysis of Map and the Tywi and Eden Catchments

Statistical analysis was performed as previously described [41,43,44]. All the rivers Tywi and Cothi catchment data including rainfall were provided by Natural Resources Wales. Briefly, a generalized linear model based approach [53] was implemented in the R statistical programming environment [54] using an analysis of variance (ANOVA) to assess the significance of river height and river flow and rainfall on the sampling day and 10 proceeding on the presence of *Map*. For each, a matrix was compiled compromising measurements of five variables, (sample date, presence/absence of *Map* IS*900*, presence/absence of *Map* F57, river flow, river height). Additionally, 11 rainfall variables were added to the matrix with rainfall on the sampling day and up to 10 days preceding the sampling date [41].

Furthermore, a linear discriminant analysis determined whether river characteristics and rainfall could be combined to form an index with which to predict the presence of IS*900* [55]. A randomization method was used to test for clustering of IS*900*-positive days [41,43]). A simple Chi-squared test of association was implemented to see whether there was a significant difference in the number of *Map*-positive samples at the Tywi upstream sampling site and the Tywi downstream (Nantgaredig Bridge) sampling site. Similarly, for the Eden Catchment, the relationship between rainfall on day 0 and up to 10 proceeding sampling with the presence of *Map* was assessed using a generalized linear model approach followed by a simple ANOVA. A chi-squared test of association was performed to see whether there was a significant difference in the number of *Map*-positive samples for each of the three sub-catchments Morland, Darce and Pow. The Eden DTC Project provided all Eden catchment data.

## 3. Results and Discussion

### 3.1. Overview

In total, 316 water samples from the River Tywi catchment were analysed over a 2-year period with 39% being qPCR positive for IS*900*. Within these IS*900*-positive samples, 13% were F57 positive. Combined weekly and monthly samples (*n* = 248; Table 1 which excluded the fine sampling data) revealed that IS*900* was detected 45% (F57, 13%), 63% (F57, 19%) and 48% (F57, 16%) in Cothi (site 6), Tywi (upstream, site 5) and Tywi (downstream, site 7), respectively (see Figure 1), with a combined detection rate of 52% (F57, 16%) over the sampling period.

Our previous study carried out 10 years ago and using a nested end-point PCR assay for IS*900* showed that the presence of *Map* in the River Tywi was predictable based on river height, river flow and rainfall up to 10 days before [41]. Returning to the River Tywi catchment 10 years later, and using a qPCR approach, this study showed that *Map* was still detectable at high rates similar to those previously observed (63% compared to 68% for the comparable sampling at site 5; Pickup et al., 2006). In addition, the significant drivers for *Map* being present in the river remain as before, namely the high Johne’s disease rate in UK cattle [38] coupled to rainfall in the catchment (this study; [41]. We can conclude (Aim 1) that presence of *Map* in the catchment had not changed over time which reflects prevalence of Johne’s disease remaining high. Whether it has changed between 2012 and the present day remains to be determined but we conclude that it would remain the same unless farming practices and stock management have changed in response to reduce the prevalence of Johne’s disease 

### 3.2. Temporal and Spatial Monitoring and Quantification of Map in Rivers

#### 3.2.1. Weekly Monitoring of Map in the River Tywi Catchment

The presence of *Map* was assessed in water within the rivers Tywi and Cothi and at their confluence on a weekly basis. The hydrography of each river differed with the River Cothi having a height range during the sampling period of 0.4 to 2.0 m and the Tywi 0.5 to 3.8 m, with flows of 0.5 to 89 m^3^ s^−1^ and 6.9 to 271 m^3^ s^−1^, respectively.

Samples were collected from the three sites sampled on a weekly basis over a 70-week period, (Figure 1) namely, River Cothi (Site 6; *n* = 61) and River Tywi upstream (site 5; *n* = 65) and downstream (which represents the confluence of the rivers Tywi and Cothi, site 7; *n* = 64) (Figure 2).

*Map* IS*900* was detected in 44% (F57, 13.1%), 62% (F57 19%) and 44% (15.6%) of river water samples collected from the River Cothi and upstream and downstream of River Tywi respectively. Detection was not continuous (Figure 2) with no detection in the River Cothi for a period of 9 weeks when the river parameters on average were 0.6 m height and 2.6 m^3^ s^−1^ but was present when the height and flow were higher (2.7 m, 17.6 m^3^ s^−1^). *Map* was absent from the River Tywi for 6 weeks in an overlapping but shorter period than found for the River Cothi, with the river height and flow of, on average, 0.8 m and 17.8 m^3^ s^−1^ respectively. Prolonged presence for a period of 8 weeks overlapped that of the Cothi with river height and flow elevating to 1.4 m and 50 m^3^ s^−1^. A further prolonged presence of 6 weeks was also associated with the height and flow of the River Tywi, of 1.8 m and 83.7 m^3^ s^−1^.

We tested the significance of river height and river flow on the presence *Map* using thresholds of 0.8 m and 8.9 m^3^ s^−1^ for the River Cothi and 1.1 m and 33.1 m^3^ s^−1^ for the River Tywi and the significance of rainfall up to 9 days prior to sampling using parameters determined previously [41] (Table 2).

For the River Cothi, the height and flow significantly indicated the presence of *Map*. In addition, rainfall preceding sampling was significant on the day before and 3 and 9 days before sampling. A similar pattern was observed upstream on the River Tywi with river height and flow significantly influencing the presence of *Map* with rainfall being significant on days 1, 2, 3 and 6 prior to sampling. The height and flow were also significant factors in the downstream sampling on the River Tywi with rainfall 1, 2 and 6 prior to sampling being significant.

In addition, Figure 3 shows that IS*900* was detected simultaneously in 34, 23 and 36% of the samples in site 5/site6, site 6/site7 and site5/site7, with 23% of the IS*900*-positive samples detected simultaneously at all sample points. F57 showed a similar pattern apart from it not being detected simultaneously at all sites. The detection rate for all three sites was not significantly different.

Overall, the presence of *Map* was confirmed to be significantly associated with river height, flow, and rainfall preceding sampling and was found to be entering the main flow from tributaries of the sub-catchments, indicating widespread *Map* deposition within the catchment.

#### 3.2.2. The Contribution of Waste Water Treatment Works (WWTWs) to the Presence of Map in Rivers

Sample sites A/B and C/D which are upstream and downstream of WWTWs on the Cothi and Tywi respectively (Figure 1) were sampled on a monthly basis for 15 months to examine whether the presence of small WWTWs increased the detection of *Map* in river water. For the Tywi, IS*900* was detected in 67% (F57, 20%) of the water samples collected upstream of the WWTW (A) and in 67% (F57, 20%) of those collected downstream of the WWTW (B) IS*900* was detected upstream and downstream on the same day on 70% of the sampling dates. *Map* cell equivalents in river water upstream and downstream of the WWTW was 1–10^4^ CE L^−1^ (IS*900*).

In the River Cothi, *Map* IS*900* was detected in 50% (F57, 29%) of river water collected upstream of the WWTW and at site D in 43% (F57, 7.1%) of river water samples collected downstream of the WWTW. *Map* cell equivalents in river water upstream and downstream of the WWTWs were estimated to be 1–10^3^ CE L^−1^ (IS*900*) and 1–10^4^ CE L^−1^ (IS*900*) respectively.

The detection rate of *Map* upstream and downstream of both WWTWs was not significantly different showing no influence over the sampling period even at low flow when *Map* from the catchment entering the rivers would be reduced (Aim 2). Unlike the River Taff, where the influence of the larger urban wastewater outflows was significant during periods of low flow, the size of these rural systems did not influence the dynamics of *Map* in the river showing for the River Tywi that the catchment field runoff was the significant factor determining the numbers of *Map* detected. We previously showed that *Map* was detected in sewage outflows from waste water treatment plants [41] and showed that the presence of *Map* in periods of low river flow were probably due to *Map* entering the river from large urban waste water treatment plants [43].

#### 3.2.3. Increased Resolution ‘Fine’ Sampling Over a Short Time Period

Half-hourly samples were taken over a 6-h period taken on three occasions from River Tywi from site 5. Additional single time point samples were taken on the same day around the catchment (River Cothi (Site 6), River Tywi (site 1) and Dulus tributaries (sites 2, 3 and 4) and at their confluence (Nantgaredig Bridge; site 7; Figure 1 and Table 3)). This sampling regime resulted in a total of 37 samples collected on three sampling dates with *Map* IS*900* being detected in 32% of the samples in the range of 1–10^3^ CE L^−1^. The detection of IS*900* at site 5 for the April sample date was 1 out of 12 (8%); for July it was 3 out of 13 (23%) with two samples being consecutive. In November, IS900 was detected in 11 out of 12 samples (92%) with three samples being positive for F57. On each occasion, five single samples were collected from within the catchment: For November, when fine sampling revealed a 91% detection rate, IS*900* was detected in all five sites compared with 60% of additional sites at a fine sampling detection rate of 23%. No additional sites were positive when the Tywi fine sampling detection rate was 8%.

#### 3.2.4. Sampling Period and Map Presence

Sampling regimes were monthly, weekly or a single day (fine sampling) with IS*900* detection rates of 38% (*n* = 58), 34% (*n* = 190) and 41% (*n* = 29), respectively, showing comparable detection rates over the three sampling regimes. Therefore, we can conclude that should *Map* need to be monitored in the future, then weekly or monthly sampling from a fixed location on a river will capture an adequate representation of the flow dynamics of *Map* in a catchment (Aim 3).

### 3.3. Quantification of Map

For the first time, the presence of *Map* was quantified from 192 of the river water samples using qPCR for IS*900* and F57 (Aim 4)*. Map* was detected in the range of 0 (not detected) to 10^5^ CE L^−1^. The samples compared with the corresponding river height and river flow at the time of sampling. This showed that as river height increased, it was significantly associated with higher numbers of *Map* for both the Tywi upstream and downstream (*p* = 5.2 × 10^−8^ and <2 × 10^−16^ respectively) and significantly associated with increased numbers of *Map* in the River Cothi (*p* = 0.0003). Due to strong collinearity between river height and river flow (correlation 0.98), quadratic discriminant analysis was applied which confirmed that higher pathogen concentrations are associated with the increase in the average river flow and river height. This would therefore account for the river carrying between 0 and 10^8^
*Map* per m^3^ of river water which was measured in the range of 0–10^5^ CE L^−1^ with the upper numbers relating period of higher rainfall which generates higher flow/height parameters for the river.

*Map* was detected in river samples using qPCR for both IS*900* and F57. F57 was only detected in samples that were IS*900* positive but, in a number of samples *Map* was only revealed by the presence of IS*900*. The relationship between the concentrations of *Map*, as measured using IS*900*, were assessed in relation to the concurrent detection of F57. Of the 192 weekly samples tested, 97 samples were IS*900* positive, of which 29 were coincidentally F57 positive (30%). The relationship between the concentration of IS*900*-positive samples and the concurrent detection of F57 was shown to be significantly associated with *Map* numbers at and above the range of 10–100 CE L^−1^, which equates to the IS*900*:F57 ratio of approximately 17:1 in *Map* [56,57]. Pickup and co-workers previously showed that *Map*, identified by IS*900,* was present in rivers in Welsh catchments at a frequency of 36% for the River Taff and 68% for the River Tywi, taken from single points over a year in 2002 [41,43]. The lower frequency detection of F57 is, as previously reported, linked to approximate 17:1 ratio of IS*900*: F57 which creates a concentration threshold below which F57 in *Map* is not detected in environmental samples.

### 3.4. Overview Map in Welsh River Catchments

We undertook three temporal regimes for sampling water of the River Tywi over monthly, weekly and every 30 min over a 7-h period with confirmatory samples taken around the area. Through the combined data of the three sampling regimes, a real time appreciation of *Map* presence was realised, showing that *Map* was regularly present in both rivers, each fed by independent catchments. Overall, *Map* was present upstream and downstream in the rivers Tywi and Cothi at the same sample time point over a period of a year in 23% of the samples, which indicated the widespread deposition of *Map* on the Tywi catchment. The Tywi downstream site at Nantgaredig Bridge represents the confluence of the two rivers, where the Tywi upstream and Cothi join. It was apparent that *Map* was commonly present in Tywi downstream and Cothi in roughly 50% of the samples taken in each river system.

The short-period high-frequency sampling on three different occasions in summer, winter and spring showed high variability driven by differences in river parameters with the average high height/flow river for the day (2.4 m/89 m^3^ s^−1^) showing 91% detection rate in 13 half-hourly samples taken in the winter compared to 32 and 8% in both sets of 12 samples taken in spring and summer when the river parameters were lower at approximately 1 m height and 24 m^3^ s^−1^. The high winter frequency may reflect higher rainfall and increased slurry spreading despite in some areas the animals being housed inside. Although *Map* was detected in each seasonal sample period not all samples within that 6-h period were positive demonstrating that even when sampling at fine resolution that patchiness of sampling with both negative and positive samples will be detected.

The different temporal sampling regimes have shown that *Map* is widespread in the catchment. It enters the catchment rivers simultaneously from different geographical locations in the same river continuum and can be detected as such by samples taken at different points on the same day within the catchment but also at times in continuously present in samples taken over a period of time. Our detection of *Map* is still considered to be an underestimate due to the variability in sampling (e.g., high frequency regime (91% vs. 8%)) coupled with limitation imposed by environmental sampling such as patchiness of samples and inhibition [43]. Despite these limitations, if monitoring schemes needed to be implemented to assess the presence of *Map* in the environment, then this study would recommend that rivers in catchments are better indicators than terrestrial-based sampling [45] and that weekly sampling at a single site, which fits many non-governmental water testing procedures, would generate an acceptable profile of *Map* in a catchment. As in previous studies, F57 is detected only in samples that are IS*900* positive and at a frequency enough to give confidence that all IS*900*-positive samples represent the detection of *Map* [43].

### 3.5. Sampling Map in New Locations

#### 3.5.1. Estuarine Samples

Sediment samples (*n* = 4) were taken in the Tywi estuary but all were negative for IS*900* (data not presented).

#### 3.5.2. Eden Catchment Field Beck Samples

A total of 47 samples from beck water and field drains were collected from Eden catchment, Cumbria, UK from February 2012 to March 2013 from three different sites within each of the three sub-catchments, Darce, Morland and Pow. (Table 4). Overall *Map* IS*900* was detected in 21% of the beck water samples with 4% being positive for F57 as well. *Map* was detected at a higher frequency in the Darce catchment (42.8%) compared to 17% and 6.6% in the Morland and Pow catchments. The presence of *Map* was shown to be significantly associated with rainfall from 7 and 10 days prior to sampling (*p* = 0.002 and *p* = 0.049 respectively). *Map* was found in the range of 0–10^3^ CE L^−1^.

*Map* distribution has already been described at a national, catchment scale and river scales [41,43,45]. Sampling of Eden catchment beck water and field drains (Cumbria, UK), although a geographically independent catchment, provided an insight into the presence of *Map* at the field scale of a catchment which has not been previously tested. The Eden catchment provides intensive grazing for cattle and sheep and *Map* was detected that the field drains and becks surrounding the fields in all three sub-catchments at frequencies probably reflected by the herd intensity in those areas but each was significantly associated with rainfall that washed *Map* from the field into the field drains and becks. Furthermore, the propensity of cattle to stand in streams and introduce chemical and faecal contamination is well documented [58,59], with *Map* also detected in stagnant streambeds in farm units in USA [59].

#### 3.5.3. Natural River Foam

Foam accumulating in the river was collected on one occasion from the River Dulas (site 4) and from the upper Tywi tributary (Site 1). Single samples were taken from two close sites on the River Dulus. One was positive for both IS*900* and F57 and the other positive for IS*900* only. This study provides the first report of *Map*, or other any pathogens, being present in natural river foam lines. Foams, although problematic in waste water treatment works [60], are ubiquitous in the environment, commonly seen as discoloured patches on streams, rivers, lakes and sea water. Although assumed to be of man-made origin as they are visually unpleasant, they are often observed in pristine environments which indicates a natural origin [47]. The stability of foams, particularly those in waste treatment works, is often associated with hydrophobic particles such as bacteria cells [60]. River foams indicate that bacteria, protozoa and/or algae are major contributors to organic matter in foams [61]. The majority of species present in foam were found to be either benthic or periphytic rather than planktonic [62]. Maynard (1968) considered foam to be an important habitat, which has been ignored previously, and it remains so. The break of foam bubbles may cause an increased risk of human exposure to pathogens [47] and provide a new route for human exposure reinforcing the observation of aerosolisation of *Map* from directly from rivers may be implicated in Crohn’s Disease clusters [43,44]. In this case, *Map* was detected in sufficient numbers in the samples to be positive for both IS*900* and F57 (>14 per sample).

## 4. Conclusions

From our aims (1–5), as stated in the introduction, we have shown that:the overall distribution and presence of *Map* in the catchment has not changed over a 10-year period and that associated farming practices and management of stock have remained the same;that the effluent small waste water treatment works does not significantly input of *Map* into the river, but large city-based waste treatment works do have a significant input;using a variety of temporal and spatial sampling regimes from monthly single sample points to fine sampling every 30 min, that weekly monitoring from a single carefully chosen location, adequately describes the presence of *Map* in rivers emerging from defined catchments or sub-catchments. Monitoring due to regulatory policy could become mandatory if *Map* is confirmed as a human pathogen and we suggest a monitoring scheme that is feasible;for the first time, *Map* concentration in rivers has been assessed and numbers ranged up to 10^8^ cell equivalents L^−1^;the detection of *Map* in dust [63], domestic showers (drinking water supplies as these supply showers) [44], river aerosols [44] and, in this study for the first time, in river foams, has extended our human exposure model further (Figure 4) [41].

Given that climate conditions and the disease status of the UK herd is unlikely to change in the foreseeable future, our model shows that *Map* exposure may continue to impact communities as previously described [43]. Furthermore, its persistence in rivers, where abstraction occurs, is likely to sustain human exposure through domestic water supply, with potential health consequences of this animal pathogen which is linked to a number a number of possible health outcomes in addition to Crohn’s disease [41,44,45]. These include Parkinson’s disease, rheumatoid arthritis, neuromyelitis optica spectrum disorder, multiple sclerosis and type 1 diabetes myelitis [4,5,6,8,9,10,11,12,64,65] and given the broad range of diseases, and the possible impact on human health, then monitoring human exposure could be crucial in disease control.

## Figures and Tables

**Figure 1 microorganisms-07-00136-f001:**
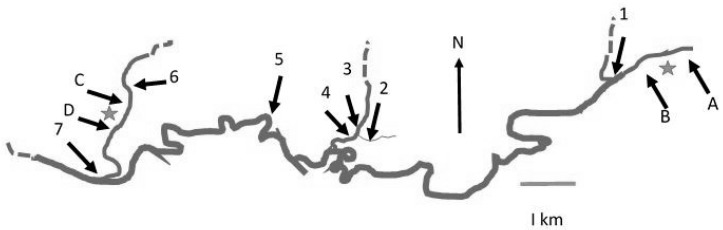
Sampling sites on the rivers Tywi, Cothi and Dulas (Sites 5, 6 and 7 represent weekly sample points; Monthly samples comprised Tywi upstream of Waste water treatment Works (WWTW *) (**A**); Tywi downstream of WWTW * (**B**); Cothi upstream of WWTW (**C**); Cothi downstream of WWTW * (**D**); Sites 1, 2, 3, 4, 6 and 7 represent sites sampled on the same day as the fine sampling site, (5) sites which were sampled on three occasions every 30 min, and the remaining sites were sampled once on each occasion).

**Figure 2 microorganisms-07-00136-f002:**
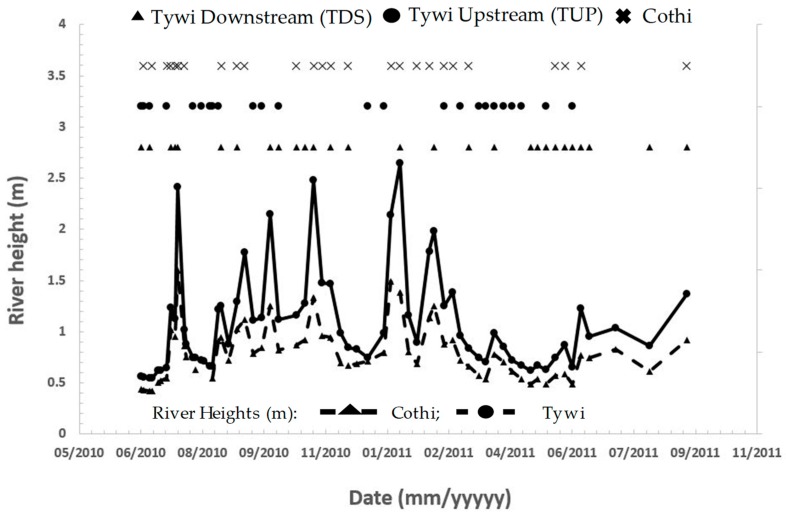
The presence of *Map* in the rivers Tywi and Cothi (Wales, UK) compared to their respective river heights (m).

**Figure 3 microorganisms-07-00136-f003:**
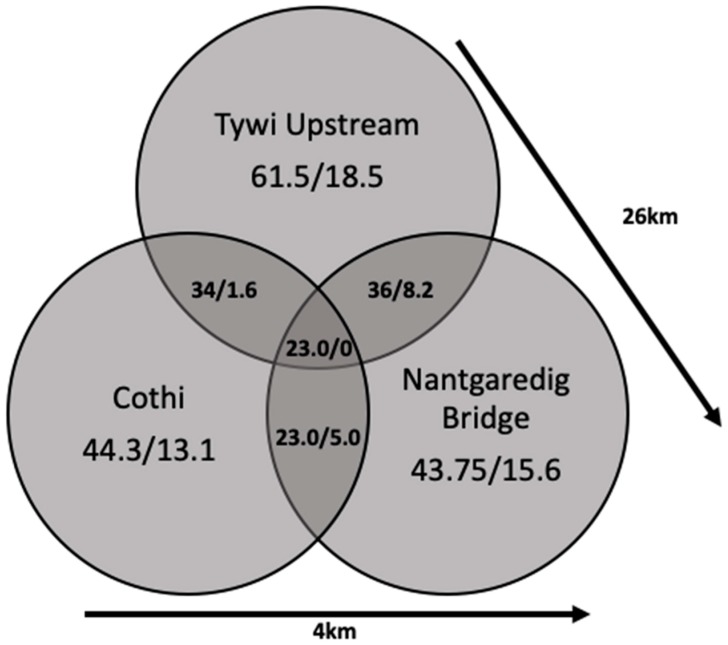
Detection of *Map* in the rivers Tywi and Cothi and at the confluence. (Arrow denotes distance from furthest sampling site (see Figure 1.) Data show detection of *Map* in individual rivers and when detected in others on the same sampling date.)

**Figure 4 microorganisms-07-00136-f004:**
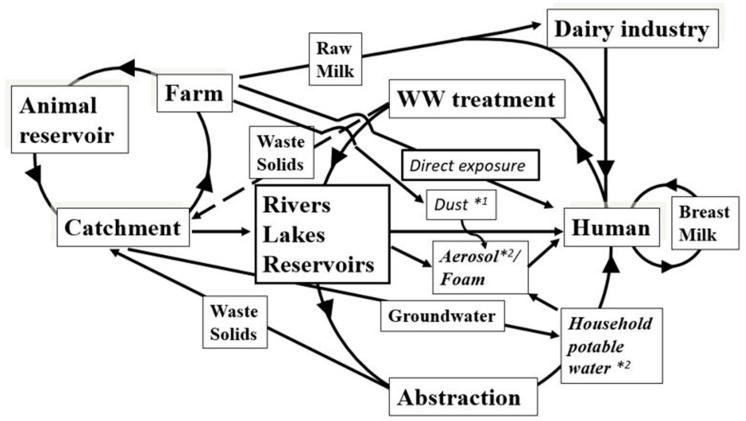
Revised model *Map* transmission and human exposure based on Pickup et al. [41] accounting for additional routes via potable water, natural foam and aerosols. *^1^ [63] *^2^ [44].

**Table 1 microorganisms-07-00136-t001:** Sampling sites for weekly/monthly samples (2010–2011).

Sample Site	UK OSGR	Sample Type	Date	Samples *n* =
Tywi upstream of WWTW * (A)	SN616212	River water	Monthly	15
Tywi downstream of WWTW * (B)	SN614211	River water	Monthly	15
Tywi upstream (5)	SN533212	River water	Weekly	65
Cothi (6)	SN505217	River water	Weekly	61
Cothi upstream of WWTW * (C)	SN506214	River water	Monthly	14
Cothi downstream of WWTW * (D)	SN502210	River water	Monthly	14
Tywi downstream (Nantgaredig; 7)	SN491203	River water	Weekly	64

Sampling sites: location and dates, (weekly/monthly samples (2010–2011); WWTW *—waste water treatment plant).

**Table 2 microorganisms-07-00136-t002:** Influence (*p* values) of river height, river flow (at the time of sampling) and rainfall (on days preceding sampling) on the presence of *Map* in rivers Tywi and Cothi. (ns: not significant).

River	Height (m)	Flow (m^3^ s^−1^)	Rainfall (Days before Sampling)				
			1	2	3	6	9
Cothi	0.0026	0.0019	0.003	ns	0.01	ns	0.018
Tywi upstream	0.00013	0.00008	0.02	0.000006	0.01	0.034	ns
Tywi downstream	0.037	0.029	0.03	0.00002	ns	0.011	ns

**Table 3 microorganisms-07-00136-t003:** Sample sites for Fine sampling (2012–2013).

Sample Site	OSGR	Sample Type	Sampling Date
Ford (1)	SN595222	Foam	24.07.2012
		Field drain water	23.04.2013
		River water	24.07/20/11.2012
		River water	23.04.2013
Dulus pre-tributary (2)	SN550212	River water	24.07/20.11.2012
		River water	23.04.2013
Dulas side tributary (3)	SN550212	River water	24.07/20.11.2012
		River water	23.04.2013
Dulas downstream (4)	SN550212	River water	24.07/20.11.2012
		River water	23.04.2013
		Sediment	23.04.2013
		Foam	24.07.2012
Tywi upstream (5)	SN533212	Sediment	24.07.2012
		Sediment	23.04.2013
		River water (FS)	24.07/20.11.2012
		River water (FS)	23.04.2013
Cothi (6)	SN505217	River water	24.07/20.11.2012
		River water	23.04.2013
Nantgaredig bridge (7) Tywi downsteam	SN494204	River water River water	24.07/20.11.2012 23.04.2013

Fine sampling (FS; 2012–2013) fine sample comprising hourly samples over a 6-h period in a day (FS-Fine sampling).

**Table 4 microorganisms-07-00136-t004:** Biweekly sampling of Field Beck Water from the Eden Catchment (2012–2013).

Sub-Catchments	Sample Site	OSRG
Morland	Long Sike	NY581196
	Sleagill Beck	NY596190
	Newby Beck	NY597212
Darce	Lowthwaite Beck	NY409236
	Thackthwaite Beck at Nabend	NY411253
	Mell Fell Beck	NY407244
Pow	Pow Beck at Beckhouse Bridge	NY422469
	Unnamed tributary Pow Beck	NY386500
	Pow Beck at Green Lane	NY386500

Biweekly sampling of Field Beck Water from the Eden Catchment (2012–2013, *n* = 13). (OSRG—UK Ordnance Survey Grid Reference).

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
