# Peer review of "Presence of Mycobacterium avium Subspecies paratuberculosis Monitored Over Varying Temporal and Spatial Scales in River Catchments: Persistent Routes for Human Exposure"

_microorganisms, 2019, doi:10.3390/microorganisms7050136_

Round 1
Reviewer 1 Report
The authors clearly showed the presence of the MAP in the environment and its effect on human health. The study show a strategy for future screening of environmental samples. I suggest to improve figure 4 by including pictures. Please also make figure 4 as the graphical abstract of the manuscript.
Author Response
Reviewer 1
I suggest to improve figure 4 by including pictures. I think the current form is more suitable to a scientific journal but if presented in a more popular form I agree pictures would be useful.
Please also make figure 4 as the graphical abstract of the manuscript. I agree and will submit it as a graphical abstract.
Reviewer 2 Report
After careful review of this manuscript I feel that the journal should accept it with minor revisions. My comments are as follows.
The manuscript by Richardson et al., provides a much needed update on the presence of Map in UK rivers. In this article, the authors have expanded on their previous research on the presence of Map in the River Tywi. Previous research demonstrated that Map was present in 69% of samples from the River Tywi as well as the positive correlation between and rain fall patterns and river flow and Map prevalence in river samples. In this article the authors sought to provide an updated model of Map prevalence and possible exposure routes in the River Tywi and its catchment areas. As Map is associated with Crohn’s disease in humans and Johne’s disease in livestock, it is extremely important that Map prevalence in the environment is closely monitored as there is a clear risk to human health.
The correlation between river height/flow and the presence of Map was clear, however this had been previously demonstrated. The authors have clearly shown that the presence of small waste water treatment works do not affect the prevalence of Map in the River Tywi and River Cothi, and that weekly or monthly sampling is adequate to monitor the prevalence of Map in these areas. Furthermore, the authors have successfully quantified Map and shown its correlation to river height and flow, however have failed to explain the significance of the lower detection of F57 in their samples compared with IS900. The authors also show a previously unknown reservoir for Map in natural river foam.
Overall, the results presented by Richardson et al., are impactful, particularly in that they demonstrate the various exposure routes of Map, and how its prevalence is predominantly determined by river height and flow. They also provide suggestions on sampling techniques for Map which will make the monitoring of Map for public health reasons more efficient and robust. The main concerns regarding the manuscript are that their discussion is inadequate to explain the importance of their results, it is suggested that the authors expand the points in their discussion to ensure each result is adequately examined. Further minor concerns are regarding the grammar and sentence structure. There are a variety of different methodologies used across a wide range of geographical areas, and I feel that these could be better organised within the text in order for the reader to acquire a clear picture of the study.
Author Response
Reviewer 2
Reviewer 2
The main concerns regarding the manuscript are that their discussion is inadequate to explain the importance of their results, it is suggested that the authors expand the points in their discussion to ensure each result is adequately examined. We agree and the importance of the results has been linked to the aims and added to the conclusion.
Further minor concerns are regarding the grammar and sentence structure. Hopefully, we have addressed these after reviewing our text. All changes shown in track changes.
There are a variety of different methodologies used across a wide range of geographical areas, and I feel that these could be better organised within the text in order for the reader to acquire a clear picture of the study. We have looked again at the text. We feel that our structure describes each geographical location in order of sampling type (same river/different types) or location (e.g Eden is separate) so changing the order we feel would lose the flow and the story. The methods are detailed in the methods section.
Reviewer 3 Report
The authors describe a study monitoring the presence of Mycobacterium avium subsp. paratuberculosis (Map) by quantitative PCR over a range of temporal and spatial scales in the river Tywi catchment. The study shows that the overall distribution and presence of Map in the catchment has not changed over a 10 year time period and that the presence of Map could be quantified and was significantly associated with river’s height, flow and rainfall preceding sampling. Additionally, effluent from small wastewater treatment plants did not significantly increase Map levels in the river water. The study data suggests that weekly or monthly sampling from a fixed location on a river will be representative of the flow dynamics of Map in a catchment area. It is also the first study to show the presence of Map in natural river foams that have the potential for aerosol transmission of the bacteria. The study is important for understanding and evaluating the potential exposure risk to humans from Map in river water given the controversial potential association between Map and Crohn’s disease.
The sampling regime is well thought through, the methods employed are appropriate and the data has been statistically analysed. There is no data presented in the report to correlate the findings with the current prevalence of Johne’s disease in livestock within the catchment area or any changes in farm management strategies.
Specific points below:
1. Line 40. The association of Map and Crohn’s disease remains a controversial issue. While the selected references suggest a significant association, others do not. Please replace “significantly” with ‘controversially’ or add that the association is controversial. Also, the use of the word “most” is controversial.
2. Line 51. “and has increased from 32 to 68% in UK’. This statement is incorrect. Three of the references quoted (28,29 and 31) refer to the same survey of the UK dairy herd carried out between 2006 and 2007 (final report in reference 31). Prevalence estimates vary according to the diagnostic test used and the statistical analyses carried out. In this survey a Bayesian modelling approach was applied to enable a single estimation of the prevalence of dairy herds in the UK that were infected with Map. The best fitting model assumed that the ELISA test was independent of the faecal pool and bulk milk PCR tests and the prevalence of Map infected herds was estimated as 34.7% (95% ci 27.6% - 42.5%). However, in the same study the prevalence estimated using serum ELISA antibody testing was 65% (95% confidence interval 56.8% - 73.4%). The prevalence did not therefore increase from 32 to 68%. Please also amend abstract line 22 accordingly.
3. Lines 81-82. There is no data in this report to show that the farming practices and management of stock remain the same. Did the authors perform a survey of farmers in the catchment area to find this out? Were veterinary records available for JD testing in the catchment area over the time period of the study to see if the number of cases of JD had increased or estimate prevalence in the catchment area? Without this data the aim can only be to determine whether the overall distribution and presence of Map in the catchment has changed over time. Please also amend abstract lines 21-22 accordingly.
4. Line 96. “It passing through are of both” should read ‘It passes through areas of both’
5. Lines 110-111. Figure 1. Legend. Abbreviation WWTW must be defined here.
6. Lines 192-197. Amend as per comments for 2 and 3 above.
7. Figure 2. Correct “mm/yyyyy” to ‘mm/yyyy’. A more descriptive figure legend would help the reader to understand this figure. The symbols are not easy to see on the graph itself.
8. Lines 234-237. Please add a sentence to explain the lower detection rates using qPCR for F57 versus qPCR for IS900 and/or refer to the discussion in section 3.3.
9. Lines 275-278. Can the authors comment more on why the sampling detection rate was so high in November compared to April and July? Presumably the higher river levels are linked to higher rainfall in November but what about the influence of farm management practices e.g. animals housed inside or manure spreading?
10. Line 316. Delete repeat of “present”
Reviewer 4 Report
The authors should support better the human health implications on detecting MAP in the environment, in particular they should cite the human diseases associated to a high positivity to MAP detection and include at least the following references in the introduction:
Anti-HERV-WEnv antibodies are correlated with seroreactivity against Mycobacterium avium subsp. paratuberculosis in children and youths at T1D risk.
Niegowska M, Wajda-Cuszlag M, Stępień-Ptak G, Trojanek J, Michałkiewicz J, Szalecki M, Sechi LA.
Sci Rep. 2019 Apr 18;9(1):6282. doi: 10.1038/s41598-019-42788-5
High levels of antibodies against PtpA and PknG secreted by Mycobacterium avium ssp. paratuberculosis are present in neuromyelitis optica spectrum disorder and multiple sclerosis patients.
Slavin YN, Bo M, Caggiu E, Sechi G, Arru G, Bach H, Sechi LA.
J Neuroimmunol. 2018 Oct 15;323:49-52. doi: 10.1016/j.jneuroim.2018.07.007. Epub 2018 Jul 21.
Mycobacterium avium subspecies paratuberculosis and myelin basic protein specific epitopes are highly recognized by sera from patients with Neuromyelitis optica spectrum disorder.
Bo M, Niegowska M, Arru G, Sechi E, Mariotto S, Mancinelli C, Farinazzo A, Alberti D, Gajofatto A, Ferrari S, Capra R, Monaco S, Sechi G, Sechi LA.
J Neuroimmunol. 2018 May 15;318:97-102. doi: 10.1016/j.jneuroim.2018.02.013. Epub 2018 Feb 28
Interferon regulatory factor 5 is a potential target of autoimmune response triggered by Epstein-barr virus and Mycobacterium avium subsp. paratuberculosis in rheumatoid arthritis: investigating a mechanism of molecular mimicry.
Bo M, Erre GL, Niegowska M, Piras M, Taras L, Longu MG, Passiu G, Sechi LA.
Clin Exp Rheumatol. 2018 May-Jun;36(3):376-381. Epub 2018 Jan 15.
The Consensus from the Mycobacterium avium ssp. paratuberculosis (MAP) Conference 2017.
Kuenstner JT, Naser S, Chamberlin W, Borody T, Graham DY, McNees A, Hermon-Taylor J, Hermon-Taylor A, Dow CT, Thayer W, Biesecker J, Collins MT, Sechi LA, Singh SV, Zhang P, Shafran I, Weg S, Telega G, Rothstein R, Oken H, Schimpff S, Bach H, Bull T, Grant I, Ellingson J, Dahmen H, Lipton J, Gupta S, Chaubey K, Singh M, Agarwal P, Kumar A, Misri J, Sohal J, Dhama K, Hemati Z, Davis W, Hier M, Aitken J, Pierce E, Parrish N, Goldberg N, Kali M, Bendre S, Agrawal G, Baldassano R, Linn P, Sweeney RW, Fecteau M, Hofstaedter C, Potula R, Timofeeva O, Geier S, John K, Zayanni N, Malaty HM, Kahlenborn C, Kravitz A, Bulfon A, Daskalopoulos G, Mitchell H, Neilan B, Timms V, Cossu D, Mameli G, Angermeier P, Jelic T, Goethe R, Juste RA, Kuenstner L.
Front Public Health. 2017 Sep 27;5:208. doi: 10.3389/fpubh.2017.00208. eCollection 2017.
Increased seroreactivity to proinsulin and homologous mycobacterial peptides in latent autoimmune diabetes in adults.
Niegowska M, Delitala A, Pes GM, Delitala G, Sechi LA.
PLoS One. 2017 May 4;12(5):e0176584. doi: 10.1371/journal.pone.0176584. eCollection 2017.
Mycobacterium avium subsp. paratuberculosis and associated risk factors for inflammatory bowel disease in Iranian patients.
Zamani S, Zali MR, Aghdaei HA, Sechi LA, Niegowska M, Caggiu E, Keshavarz R, Mosavari N, Feizabadi MM.
Gut Pathog. 2017 Jan 3;9:1. doi: 10.1186/s13099-016-0151-z. eCollection 2017.
Serum BAFF levels, Methypredsinolone therapy, Epstein-Barr Virus and Mycobacterium avium subsp. paratuberculosis infection in Multiple Sclerosis patients.
Mameli G, Cocco E, Frau J, Arru G, Caggiu E, Marrosu MG, Sechi LA.
Sci Rep. 2016 Jul 7;6:29268. doi: 10.1038/srep29268.
Is there a role for Mycobacterium avium subspecies paratuberculosis in Parkinson's disease?
Arru G, Caggiu E, Paulus K, Sechi GP, Mameli G, Sechi LA.
J Neuroimmunol. 2016 Apr 15;293:86-90. doi: 10.1016/j.jneuroim.2016.02.016. Epub 2016 Feb 27.
Author Response
We have included all of the references suggested into our text.